# The Anti-Inflammatory Effect of Bovine Bone-Gelatin-Derived Peptides in LPS-Induced RAW264.7 Macrophages Cells and Dextran Sulfate Sodium-Induced C57BL/6 Mice

**DOI:** 10.3390/nu14071479

**Published:** 2022-04-01

**Authors:** Lujuan Xing, Lijuan Fu, Songmin Cao, Yantao Yin, Lanlan Wei, Wangang Zhang

**Affiliations:** 1Key Laboratory of Meat Processing and Quality Control, MOE, Jiangsu Synergetic Innovation Center of Meat Processing and Quality Control, Nanjing Agricultural University, Nanjing 210095, China; lujuanxing@njau.edu.cn (L.X.); 2019108068@njau.edu.cn (L.F.); yantaoyin@126.com (Y.Y.); 2School of Food and Wine, Ningxia University, Yinchuan 750021, China; songmin_cao@126.com; 3College of Food Engineering, Anhui Science and Technology University, Fengyang 233100, China; w13851665574@126.com

**Keywords:** bovine bone gelatin, peptides, inflammatory cytokines, colon, gut microbiota

## Abstract

The bioactive peptides hydrolyzed from bone collagen have been found to possess health-promoting effects by regulating chronic diseases such as arthritis and hypertension. In the current study, the anti-inflammatory effect of bovine bone gelatin peptides (GP) was evaluated in 264.7 macrophages cells and followed by animal trials to investigate their interference on inflammatory cytokines and gut microbiota compositions in dextran sodium sulfate (DSS)-induced C57BL/6 mice. The GP was demonstrated to alleviate the extra secretion of interleukin-6 (IL-6), nitric oxide (NO) and tumor necrosis factor-α(TNF-α) in lipopolysaccharide (LPS)-induced RAW264.7 cells. In DSS-induced colitis mice, the gavage of GP was demonstrated to ameliorate the IBD symptoms of weight loss, hematochezia and inflammatory infiltration in intestinal tissues. In serum, the proinflammatory cytokines (TNF-α,IL-6, MCP-1, IL-1β) were suppressed along with the decreasing effect on toll-like receptor 4 and cyclooxygenase-2 by GP treatment. In the analysis of gut microbiota, the GP was checked to modulate the abundance of *Akkermansia*, *Parasutterella*, *Peptococcus, Bifidobacterium* and *Saccharibacteria*. The above results imply that GP could attenuate DSS-induced colitis by suppressing the inflammatory cytokines and regulating the gut microbiota.

## 1. Introduction

Certain widespread chronic diseases, including type 2 diabetes, atherosclerosis, and inflammatory bowel disease (IBD), have been reported to be associated with the generation of pathophysiologically inflammatory responses [1]. IBD is classified as a chronic inflammatory disease and is the most widespread metabolic disorder, affecting intestinal health all over the world [2]. During the last 20 years, the IBD incidence rate is still rising according to worldwide epidemiologic research [3]. Common therapy to reduce intestinal inflammation relies on anti-inflammatory drugs including steroidal and nonsteroidal drugs. However, the drugs may also inspire side effects of gastrointestinal irritation and increase the risk of cardiovascular and gastrointestinal bleeding [4]. Thus, the development of novel candidates from natural components is of great urgency for improving intestinal health. 

As a source of luminal components, dietary ingredients such as proteins, peptides, and amino acids are regarded as beneficial molecules for maintaining the intestinal environmental balance and promoting the remission effect on IBD patients. The bioactive compounds including glutathione, γ-glutamyl dipeptides, and amino acids have been reported to exhibit their anti-inflammatory and antioxidant effect in vitro and also modulate the immune system and intestinal oxidative stress in animal trials [5,6]. The bioactive peptides in bovine whey [7], oyster soft tissue [8], and sturgeon muscle [9] were all demonstrated to have the anti-inflammatory effect on suppressing the expression of the inflammatory cytokines in RAW264.7 macrophages cells. Thus, the RAW264.7 macrophages were generally used as a cell model to evaluate the anti-inflammatory effect of bioactive compounds as its sensitivity to immune stimulation. Isolated from egg white proteins, MLGATSL, DEDTQAMPFR, DEDTQAMPF, and MSYSAGF significantly inhibited the IL-8 and TNF-α secretion and also down-regulated the mRNA expression of TNF-α, IL-17, IL-6, MCP-1, IL-1β, and IFN-γ in colitis mice [10]. Besides, the gut microbiota plays an important role in the homeostasis of intestinal health, meanwhile, diet nutrition could induce the changes of microbiota composition as well as their biological functions, which finally posed the bidirectional communications among the intestinal microbiota and the host [11]. Similar to a previous report, the intake of proteins or bioactive peptides would regulate the composition of gut microbiota, which in turn could participate in the immune responses as well as mucosal transport system [12]. These studies all implied that dietary protein hydrolysate or bioactive peptides could get involved in regulating the composition of the microbiota, which then subsequently affect the immune system and gut homeostasis in promoting gut health. 

In the meat livestock slaughtering industry, bone is regarded as a by-product with further processing needed for use in feed and fertilizer. Besides water, protein is the major composition, accounting for 11–15% of bone among which collagen accounts for 40–50% of total proteins [13]. Collagen could be further produced to protein hydrolysate as well as bioactive peptides with beneficial effects, such as antihypertension, antioxidation, immune-regulating effect, and prevention of osteoarticular diseases [14,15]. Chicken bone-collagen-derived peptides significantly suppressed the secretion of IL-1β and TNF-α in cartilage cells, where the small molecule weight fractions showed stronger free radicals scavenging and anti-inflammatory effect [16]. In addition, the collagen-derived peptides were also demonstrated to relieve arthritis and rheumatoid arthritis symptoms [17]. Till now, few studies have ever focused on the regulation effect of collagen-derived peptides on intestinal inflammation and gut microbiota. Therefore, the inflammatory regulating effect of bovine bone-gelatin-derived peptides (GP) was tested in the RAW264.7 macrophage cells in vitro. Based on this, the physiological regulatory activity of GP was evaluated in DSS-induced mice colitis, with the hope to interpret their beneficial effect on gut health.

## 2. Materials and Methods

### 2.1. Materials

Fetal bovine serum (FBS) was purchased from KeyGEN Biotech Co. Dulbecco’s modified Eagle cell culture medium (DMEM) was purchased from Grace Biotechnology (Nanjing, China). qPCR Master Mix was purchased from Vazyme (Nanjing, China). CCK-8 kit was purchased from Biyotime Biotech Co. LPS, Cocktail protease inhibitors and Alcalase (2.4 L) were all obtained from Sigma-Aldrich Co. (St. Louis, MO, USA). Dextran sodium sulfate (MW: 36,000–50,000) was purchased from YEASEN Biotechnology (Shanghai, China). RIPA Lysate and Bradford protein quantitative kits were purchased from Beyotime Biotech Inc. (Shanghai, China). TNF-α, IL-6, MCP-1, IL-1β, and LPS ELISA kits were purchased from Neobioscience Biological Technology (Shenzhen, China). All other chemicals were of analytical grade.

### 2.2. Preparation of GP

Bovine bone gelatin was prepared in the laboratory as reported by Cao, et al. [15]. After dissolving the bovine bone gelatin in deionized water with the final concentration of 80 mg/mL, the pH of the solution was then adjusted to 9.0 by NaOH (2 M). The solution was heated at 55 °C along with adding Alcalase to hydrolyze the gelatin protein for 6.5 h. Afterward, the gelatin hydrolysate was heated at 100 °C for 20 min to stop the enzymatic hydrolysis process. After cooled to 25 °C, the pH of gelatin hydrolysate was adjusted to 7.0 by HCl (2 M) and followed by centrifuging to remove the sediment. The supernatant was then purified by Molecular membrane (3 kDa, Millipore, Billerica, MA, USA) to collect the fraction of <3 kDa gelatin peptides. The composition of peptides was analyzed by LC-MS/MS as shown in Appendix A.

### 2.3. The Cell Viability and Cytokines Secretion in RAW264.7 Cells

RAW264.7 cells were seeded in DMEM (containing 10% FBS and 1% penicillin/streptomycin) medium under 37 ℃and a 5% CO_2_ incubator. The GP concentrations were adjusted with DMEM to 0.1, 0.3, 0.5, 0.7, 1.0, 1.5 and 2.0 mg/mL respectively. In the cell viability testing, the cells were cultured in 96-wells plates with the density of 1 × 10^5^/well and then treated by GP (100 μL/wells). After 12 h, the CCK-8 solution was added with 10 μL/well and cultured for 2 h. Then, the absorbance was measured at 470 nm. For the blank group, the DMEM medium was mixed with CCK-8 only. Normal control represented the cells without GP incubation whereas the GP groups represented peptide and CCK-8 together. 

Cell survival rate (%) = OD_GP_/ OD_NC_ × 100. Here, the OD_GP_ and OD_NC_ values are the observed values minus the blank group. 

For the anti-inflammatory activity of GP, the RAW264.7 cells were seeded in 48-well plates and cultured for 24 h until reaching the density of 70–80%. Afterward, the DMEM medium was replaced by GP solution (0.1, 0.5, and 1 mg/mL) to be preincubated for 1 h. In the positive control (PC) group, the LPS was supplemented with the final concentration of 1 μg/mL for an additional 12 h. In the normal control (NC) group, the cells were cultured in a DMEM medium only without GP or LPS. After treatment for 13 h, the cell supernatant and cell lysis were all collected to measure the concentration as well as mRNA expressions of cytokines. 

### 2.4. The DSS Induced Mice Trial

A total of 32 healthy male C57BL/6 mice (10 weeks old) were purchased from the Model Animal Research Center of Nanjing University (Nanjing, China, SCXK<Jiangsu>20180027). All mice were maintained in Animal Center of Nanjing Agricultural University (SYXK<Jiangsu>2011-0037) under the standard light–dark cycle home (25 °C, humidity of 50%) and supplied with normal diet (D12450K, protein 20%, carbohydrate 70% and fat 10%) and purified water. Following the regulations of the National Guidelines for Experimental Animal Welfare and Ethical Committee of Experimental Animal Center of Nanjing Agricultural University, the mice trials were carried on. After 7 days of adaptation, the mice were weighed (bodyweight 22.01 ± 0.51 g) and randomly divided into four groups including negative control (NC), positive control (PC), GP-treated group 1 (GP-1, 100 mg/kg/day) and group 2 (GP-2, 300 mg/kg/day). According to the study of Chen et al. [18], the colitis mice model was induced by drinking DSS water (1.5%, *w*/*v*, YEASEN Biotechnology, Shanghai, China) for 10 days. For the NC group, the regular diet and purified water were prepared for the whole process. In the PC group, the DSS (1.5%) water was supplied to mice in the first 10 days, and then administered with purified water by intragastric gavage for additional 10 days. In the GP group, the DSS water was supplied the same with PC in the first 10 days, after that, the GP was supplied to mice with 100 or 300 mg/kg/day in GP-1 and GP-2 groups, respectively. During the trial, the body weight, the feces, and the diarrhea condition were recorded every two days until the mice were sacrificed by cervical dislocation at the end of the experiment (20 days). Colon tissue was collected from all animals, the colon length was measured and the fresh feces was collected into sterile tubes. In addition, the fecal samples were collected for intestinal microbe sequencing.

### 2.5. Evaluation of Disease Activity Index

Mice were weighed and recorded for weight and fecal viscosity, fecal retention or bleeding, and appearance were checked every 2 days. The grade of intestinal inflammation activity was assessed using the disease activity index (DAI). Clinical scores were evaluated as follows: stool score: 0 = normal; 1 = wet / sticky stool; 2 = soft stool; 3 = diarrhea. Bloody stool: 0 = no blood; 1 = stool or perianal blood; 2 = severe bleeding. Appearance performance: 0 = normal; 1 = fur fold or gait change; 2 = lethargy or dying state. At the end of the animal trials, the length of colons was recorded for every single mouse.

### 2.6. Biomarkers in Serum and Colon

The levels of TNF-α, IL-1β, IL-6, MCP-1, and LPS in serum were determined according to the instructions of the ELISA kit (Neobioscience Biological Technology, Shenzhen, China). The mRNA levels of genes (TNF-α, cyclooxygenase-2, Toll-like receptor 4, MCP-1, ZO-1, and Occludin) involved in inflammation and tight junction were determined by quantitative real-time reverse-transcription PCR (qRT-PCR) according to the method of Zhai, et al. [19]. The colon tissue was intercepted and homogenized in Trizol (1 mL) together with grinding beads. Then the total RNA was extracted by Cell RNAprep pure Kit and quantified by NanoDrop Quick Tester (Thermo Fisher Scientific, Waltham, MA, USA). Afterward, the cDNA was synthesized according to the PrimeScript RT reagent Kit instructions. In the PCR amplification reaction, the reaction volume was 20 µL, where the 8.8 µL of cDNA was mixed with 11.2 µL of a preformed mix (10 µL TB green premix EX Taq II, 0.4 µL 50×ROX reference, 0.4 µL forward primer and reverse primer respectively). As shown in Appendix A, the primer sequences were synthesized in the Gen Script (Nanjing, China). The PCR amplification reaction conditions were as follows: denaturation at 95 °C for 15 s, annealing at 60 °C for 1 min, 45 cycles, and determination of CT value. The expression of GAPDH was used as the reference housekeeping gene. The change times of gene expression levels were calculated by a 2^−ΔΔCt^ method. 

### 2.7. Western Blot

The proteins in mice colon were extracted by RIPA lysis buffer with the supplement of protease–phosphatase inhibitors (1%, Solarbio, Beijing, China) and then homogenized by the tissue-grinding machine with liquid nitrogen. After 15,000× *g*, 4 °C for 10 min, the proteins in the supernatant were collected and then tested by BCA protein assay reagent (Thermo Scientific, Shanghai, China). After mixing with loading buffer, the proteins were boiled for 5 min. In the running process of Western blot, the proteins were firstly separated by electrophoresis with the 4–12% gel and then transferred onto the polyvinylidene difluoride membranes (PVDF). After that, the PVDF membranes were sealed with 5% nonfat milk for 1 h and then incubated with primary antibodies for 12 h at 4 °C. The primary antibodies include rabbit anti-TLR-4 (1:1000, Abclonal, Wuhan, Hubei, China), rabbit anti-Occludin (1:1000, Abclonal, Wuhan, Hubei, China), rabbit anti-COX-2 (1:2000, Abclonal, Wuhan, Hubei, China), rabbit anti-β-actin (1:5000, Sigma, St. Louis, MO, USA). After incubation overnight, the PVDF membranes were washed by tris-buffered saline tween (TBST) solution 3 times and followed by incubation with HRP-labeled secondary antibody (goat antirabbit antibody, 1:5000, Abclonal, Wuhan, Hubei, China) at room temperature for 2 h. After washing, the membranes were incubated with ECL solution (Millipore Corporation, Billerica, MA, USA). The β-actin was used as referenced protein. Finally, the membranes were scanned by Image Quant LAS 4000 (GE, Fairfield, CT, USA).

### 2.8. Gut Microbiota Analysis

According to the study of Chen, Xie, Dai, Peng and Yi, [18], feces was collected before the sacrifice of the animals and the total DNA of feces was extracted by FastDNA SPIN Kit. For 16S rRNA amplicon sequencing, DNA samples were analyzed by Genesky Biotechnologies (Shanghai, China). Briefly, the integrity of isolated genomic DNA was tested by agarose gel electrophoresis, and afterward, the quality of genomic DNA was measured by Nanodrop 2000 (Thermo Scientific, Shanghai, China). In the V3-V4 hypervariable regions of the 16S rRNA gene analysis, the primers of 341F (5′-CCTACGGGNGGCWGCAG-3′) and 805R (5′-GACTACHVGGGTATCTAATCC-3′) were used. The Illumina NovaSeq 6000 sequencer (Illumina, CA, USA) was subsequently used for the sequence reactions process. Raw sequence data statistics were performed by Quantitative insights into microbial ecology (QIIME2) along with the quality control by the DADA2 plugin. Based on the recognition rate of 97%, the sequences were classified into actionable taxonomic units. Samples cluster trees were performed by the R Vegan package (V3.5.0). Community composition analysis, diversity analysis, as well as functional prediction analysis were performed using the cloud analysis platform (http://www.geneskybiotech.com, accessed on 18 Noverber 2021). 

### 2.9. Statistical Analysis

Data are represented as the mean ± standard deviation (SD). The data analysis was performed by GraphPad Prism version 5.0 (San Diego, CA, USA) depending on the one-way ANOVA with Dunnett’s test. The abundances of OTUs were analyzed by the R Vegan package (V3.5.0) with Tukey’s honest significant difference (HSD) test. A *p*-value of <0.05 represented significant differences. 

## 3. Results

### 3.1. Anti-Inflammatory Effect of GP in RAW264.7 Cells

There were 1434 peptides identified in the GP with the molecule of 987.39 Da to 2317.11 Da and nonapeptide accounted for 20%. Most of the peptides were composed by 7-15 amino acids as shown in Figure 1. The peptides’ sequences, mass, *m*/*z*, and accession proteins information are shown in Appendix A. In the treatment of RAW264.7 cells, concentration of 0.1, 0.3, 0.5, 0.7, 1.0, 1.5 and 2.0 mg/mL GP showed no significant effect on cell viability (Appendix A). In the study of Gao et al. [9], cell viability was significantly increased after being treated with 0.125–1.0 mg/mL sturgeon muscle peptides. However, the 2.0 mg/mL treatment decreased cell viability from 100% (NC) to 91%. The decreasing effect of peptides on cell viability may be related to the increased osmotic pressure, which even causes the death of cells. Currently, the concentration of 0.1–2.0 mg/mL GP was measured to be no toxicity on macrophages cells. In the LPS induced RAW264.7 cells, the GP treatment suppressed the inflammatory biomarkers as shown in Figure 2. Compared with PC, the pretreatment of GP inhibited the secretion of NO in 0.1, 0.5 and 1.0 mg/mL (*p* < 0.05) and there were no differences among GP groups (Figure 2A, *p* > 0.05). Similarly, the GP pre-treatment also suppressed the TNF-α from the 2361 pg/mL in PC to be 1498 pg/mL in 1.0 mg/mL of GP groups (Figure 2B, *p* < 0.05). Here, the 0.5 and 1.0 GP groups showed no differences from each other. To reflect the anti-inflammatory effect of GP, the cytokine mRNA expression was also measured by RT-PCR with the groups of 0.5 and 1.0 mg/mL, respectively. Compared with PC, the GP reduced TNF-α mRNA expression by 1.79- and 2.29-times in 0.5 and 1.0 mg/mL groups (Figure 2C, *p* < 0.05). In addition, GP also reduced IL-6 mRNA expression compared with the PC group (Figure 2D, *p* < 0.05). The 0.5 mg/mL of GP group showed no differences in IL-1β expression compared with PC (*p* > 0.05), whereas the 1.0 mg/mL of GP group was investigated to exert the suppressing effect on IL-1β (Figure 2E, *p* < 0.05). Regarded as the potential resource of anti-inflammatory components, the dry-cured ham peptides were also checked to repress the proinflammatory mediators of TNF-α, IL-6, IL-8, and IL-1β in the LPS-induced RAW264.7 macrophage cells, where the isolated sequences of GPPGL, GPAGPL, and GPPGAP were all originated from collagen protein [20]. Derived from dynastid beetle, the bioactive peptides inhibited the cyclooxygenase-2 (COX-2), iNOS, IL-6, and IL-1β expression [21]. Isolated from microalgae protein, the peptides with the MW of <3 kDa were also investigated to suppress the TNF-α, COX-2, and IL-6 expression in RAW264.7 macrophage cells [22]. The LPS-induced RAW264.7 macrophage cells were widely used to assess the anti-inflammatory effect of bioactive peptides and the suppressing effect on proinflammatory cytokines was the biomarker to reveal the immune level in cells. From the current results, the GP sequences were mainly composed by 7-15 amino acids and the GPP fragment appeared in 397 peptides such as FLPAGPPE, GRGPPLGF, GPPGFQGPK, GPPGFGPGY, GPPGEPGPQ, GPPGFPGLE, etc. Similarly, the anti-inflammatory peptides in dry-cured hams (GPPGL, GPPGAP) was also composed by GPP fragment, which means the certain sequences play an important role in inflammation function [20].

### 3.2. GP Attenuated DSS-Induced Acute Colitis Symptoms

Several studies have reported that the DSS-induced mice would have the symptoms of weight loss, colon shortening, as well as hematochezia [23]. As revealed in Figure 3A, the mice trials were separated with NC, PC, GP-1, and GP-2 groups. In the NC group, the body weight of mice was increased gradually and reached 25.8 g at the end of the trial of Day 20 (Figure 3B). Compared with NC, the DSS treatment inspired significant body-weight loss in the PC and GP group during the former 10 days. Starting from day 4, the body weight decreased gradually until day 12. As expected, the GP treatments could attenuate body weight losing tendency compared with the PC group. In special, diarrhea and stool bleeding symptoms were also revealed the IBD condition according to the DAI scores (Appendix A). After the GP treatment, the DAI values were alleviated compared with PC. Known from the length of the colon, the DSS treatment was shown to shorten the colon from 8.23 cm (NC) to 7.65 cm (PC), whereas the GP treatment could recover the colon length compared with PC (Figure 3C,D; *p* < 0.05). 

Without the inflammatory stimulation, the colon showed intact colonic mucosa, crypts, stroma, and submucosa, and few inflammatory cells infiltrated in submucosa and ulceration in NC (Figure 4A). After DSS treatment, the mice showed intense inflammatory lesions, including the hyperplasia of crypts, the increased macroscopic spaces between crypts, severe epithelial cells damage, submucosal edema, and inflammatory cellular infiltration in the submucosa, which were similar to the previous report on the DSS-induced colitis model [11]. However, the treatments of GP ameliorated the inflammatory symptoms in the colon, including relative intact surface epithelium, lower infiltration of inflammatory cells, mild submucosal edema, and crypt glands close to NC. According to the histopathological score (Figure 4B), the PC groups showed much higher scores than NC (*p* < 0.05), and the pretreatment of GP decreased histopathological score as being expected (*p* < 0.05). In the study of Wan et al. [11], the body weight of mice was decreased after 8 days of DSS treatment. Depending on the DAI score, the body weight and the colon length were all alleviated in the supplement of dicaffeoylquinic acids revealing the anti-inflammatory effect of dicaffeoylquinic acids in IBD mice. The intake of γ-Glutamyl cysteine peptides (γ-EC, 150 mg/kg) was also checked to suppress the IBD symptoms of reducing weight loss, stool blood, and diarrhea levels in colitis mice. Where the γ-EC participated in the regulation of calcium-sensitive receptors along with the regulation on c-Jun N-terminal kinase (JNK) and nuclear factor kappa-B (NF-κB) pathways, thus playing a role in ameliorating the inflammatory symptoms of IBD [6]. Inconsistent with previous studies, the typical symptoms of weight loss, mucosal inflammation as well as hematochezia pathological changes were all generated in the DSS-treated mice, meanwhile, the supplement of GP ameliorated the IBD-related symptoms. During the digestive absorption, the structure of peptides would be changed by digestive enzymes in the gut; however, there were still some peptides that could be transferred by the paracellular route transport through the intestinal epithelial cells and then exhibiting their anti-inflammatory activities. 

### 3.3. Elisa and mRNA Analysis of Inflammatory Cytokines

Besides the tissue damage in the histopathological image, the colitis is generally accompanied by the abnormal secretion of proinflammatory cytokines as well as immune chemokines [24] As showed in Figure 5, the stimulation of DSS significantly increased the levels of TNF-α, IL-6, IL-1β, MCP-1 and LPS than NC group (*p* < 0.05), whereas the supplementation of GP notably inhibited the DSS-induced production of TNF-α, IL-6, IL-1β, MCP-1, and LPS as compared with PC. For instance, the TNF-α content in serum increased from 411 pg/mL (NC) to 1021 pg/mL (PC), whereas the pretreatment with 300 mg/kg/day (GP-2) significantly inhibited the TNF-α content to be 403 pg/mL. Simultaneously, the content of IL-6 and MCP-1 in serum was also suppressed in GP-2 group, and the dose of 100 mg/kg/day in GP-1 showed no significant differences compared with PC (*p* > 0.05). According to the inflammatory cytokines in serum, the gavage of GP had a dose manner with the effective dosage of 300 mg/kg/day. Compared with NC, the stimulation of DSS also increased the mRNA expression of inflammatory cytokines, such as TNF-α, COX-2, and MCP-1, as shown in Figure 6. In GP groups, the aberrant secretion of biomarkers was significantly alleviated as compared with PC (*p* < 0.05). For instance, the expression of TNF-α in GP-2 group was decreased than PC, meanwhile, the COX-2 and Toll-like receptor 4 (TLR-4) was also decreased respectively (*p* < 0.05). According to the protein expressions in Figure 7, the COX-2 was also demonstrated to be increased in PC and the supplement of GP showed to have a decreasing effect without significant differences. Indeed, the TLR-4 showed a same tendency as revealed in RT-PCR, where the PC groups expressed higher content than NC, and the GP treatment exhibited a suppressing effect on TLR-4. During the formation of IBD, TNF-αplays a remarkable role in the pathogenesis, which could recruit more inflammatory factors to infiltrate the intestinal tissue and result in intestinal tissue lesions and inflammatory responses [25]. Thus, the anti-TNF-α component has also been reported as a possible therapy to reduce the pathology in IBD patients. Currently, the GP-2 was measured to suppress the TNF-α secretion with the intake of 300 mg/kg, implying a significant role in the IBD. 

Simultaneously, the DSS stimulation also inhibited the mRNA expression of tight junction proteins occludin and ZO-1 (*p* < 0.05), which is consistent in the histological image along with the DSS-induced intestinal mucosal damage. In the GP group, the ZO-1 and occludin mRNA expression was recovered than PC, implying the protective effect of GP on the mucosal tissue. As revealed in Figure 7C, the expression of occludin was also demonstrated to be improved compared with PC group (*p* < 0.05). Tight junction proteins played a major component for the intestinal tract barrier with the function of controlling the paracellular spaces as well as mucosal tightness. The intercellular bridge is established by multiple proteins such as occludin, junctional adhesion molecules, claudins, ZO-1, and others. In special, the ZO-1 is the main linkage between occludin and other apical prejunctional proteins [26]. The peptides from mucosal protein were investigated to induce the extent of the mucosal barrier as well as pathological features in DSS-induced colitis mice, where the ZO-1 and occludin proteins were enhanced compared with that of DSS group [27]. In addition, the administration of mucosal-derived peptides could also reduce cell detachment, necrosis in the mucosal damage site along with promoting cell proliferation. The multifunction on the colonic mucosal barrier as well as the beneficial effects in colonic inflammation supply the potential of using the peptide as the therapeutic agent on the IBD patients. Similarly, the promoting effect on the ZO-1 and occludin expressions in GP groups implies their protective capacity on the tight junction and intestinal barrier, which is the mitigative reflection on the IBD symptoms. 

### 3.4. Effect of GP on Gut Microbiota

#### 3.4.1. The Structure of Gut Microbiota

As the largest and most complex microecosystem of organs, the gut microbiota plays important physiological functions on immunity, nutrition, and metabolism [11]. Thus, the dysbiosis of gut microbiota would also inspire serious metabolic diseases such as IBD, obesity, and malnutrition [28]. In the gut microbiota structural comparison among IBD patients and healthy individuals, the diversities were significantly observed in special of *Porphyromonadaceae*, *Helicobacter*, *Parasutterella*, *Parabacteroides*, *Oscillibacter*, and *Firmicutes* [11,28]. In the current study, there were 1,283,475 clean reads observed in the total 32 feces samples, which possessed average reads of 40,104 for each sample. At the species level, 1143 distinct OTUs were investigated under the sequence similarity of 97%. According to the Shannon Rarefaction and curves (Appendix A), the plateau was reached gradually along with the coverage of 99%, which implied that a reasonable diversity was captured with adequate sequencing depth. As shown in Figure 8, the cluster tree, PCA, and the PCoA analysis were displayed to reveal the similarities of microbial communities. The distinct position of NC and PC groups demonstrated that the microbial composition was separated to be different branches. There was some overlap between GP-1 and GP-2 groups, indicating that the similarities of microbial communities were much higher among them. As shown in Figure 8B,C, the variation of PC1 and PCo1 accounted for 64.66% and 34.63% respectively, implying a significant difference between NC and PC groups. In addition, the results of PLS revealed that GP-1 and GP-2 groups had a higher similarity, which exhibited obvious differences with the PC group. These results indicate that the supplementation of GP would impact the structure of gut microbiota in DSS-treated mice.

#### 3.4.2. The Phylum, Family, and OTU Level of Gut Microbiota

As shown in stacked histograms (Figure 9A), the Firmicutes, Actinobacteria, Bacteroidetes, Verrucomicrobia, Proteobacteria, Deferribacteres, and Tenericutes, were identified, in special, Firmicutes, Bacteroidetes, and Verrucomicrobia were the main phylum. The DSS stimulation was investigated to reduce the relative abundances of *Betaproteobacteria*, *Deltaproteobacteria*, *Bacilli*, *Mollicutes*, *Erysipelotrichia*, *Deferribacteres* and increased the abundances of *Verrucomicrobiae* and *Deferribacteres*. In the group of GP intervention, the relevant abundant of *Betaproteobacteria*, *Bacilli*, and *Mollicutes* could be reversed compared with the DSS-induced group. However, the abundance of *Verrucomicrobiae*, *Erysipelotrichia*, and *Deltaproteobacteria* showed no significant differences with PC. At the genus level, the *Akkermansia*, *Allobaculum*, *Bacteroides*, *Peptococcus*, *Bifidobacterium*, *Parasutterella*, *Saccharibacteria*, and *Clostridium* were identified, among which, *Akkermansia*, *Allobaculum*, *Bacteroides* were the main genus in all groups. The *Akkermansia* is demonstrated to be a promising probiotic candidate for gut health and a decreasing abundance is revealed in IBD patients [29]. In the comparison between NC and PC group (Figure 9B), the *Akkermansia* also had significant differences where the DSS (18.7) posed to have lower abundance than NC (26.8%). As expected, the two-dose GP intervention improved the *Akkermansia* abundance and there were no differences between GP-1 (32.2%) and GP-2 (34.5%, *p* < 0.05). In addition, the DSS stimulation was investigated to decrease the relative abundances of *Parasutterella* from NC (3.1%) to 0.20% (*p* < 0.05), and the two-dose GP supplement had an improving effect with the abundance of 0.9% and 1.1%, respectively.

Based on the taxonomic composition, the gut-microbiota-induced metabolism function was compared in PC- and GP-treated groups. As shown in Appendix A, the stimulation of DSS caused a down-regulation of genes involved in protein export, peroxisome, lysine degradation, carbon fixation, folate biosynthesis, nitrogen metabolism, lipopolysaccharide biosynthesis, biotin metabolism, as well as RNA degradation. The genes related to the bioactive process of ansamycins biosynthesis, glycerolipid metabolism, lysine biosynthesis, pentose phosphate pathway, and phosphotransferase system were all up-regulated in PC. Produced from pathogens and commensal bacterial, the toxins would be generated by the ansamycins and lipopolysaccharide biosynthesis, which inspired the clinical intestinal inflammation in the host tissues with the damage on the immune system [30]. 

## 4. Discussion

It has been reported that DSS-induced colitis in mice had weight loss, diarrhea, hematochezia, and colon shortening symptoms. In the present study, the mice were checked for significant blood and diarrhea in the feces after 10 days of DSS treatment. As expected, the supplement of GP in two dosages could increase the weight loss along with the relieving effect on diarrhea and blood in the feces. As revealed in the scores of DAI, treatment with GP had a decreasing efficacy on the IBD symptoms. In addition, the abnormalities of inflammatory cytokines in serum, such as TNF-α, IL-6, IL-1β, and MCP-1, also played a vital role in the formation of IBD [31]. Compared with the DSS-induced group, the supplement of GP decreased secretion of inflammatory cytokines in serum. It is well known that the LPS is produced by Gram-negative bacteria with the effect of stimulating epithelial cells to increase COX-2 and then produce the intestinal inflammatory cascade [32]. Thus, the COX-2 can be regarded as a proinflammatory mediator and also a novel target participating in inflammation. In the current study, the change of COX-2 was investigated by expressions at the mRNA and protein levels, where the DSS induced a higher COX-2 expression and the GP had the relieving effect. TLR-4 is a signal of activating the NF-κB pathway with the trigger of further inflammatory cytokines secretion in IBD [33]. As been demonstrated in mRNA and protein levels, the TLR-4 was found to be elevated in PC and be further suppressed by GP treatment. In general, the regulation of these inflammatory factors is significant proof for the treatment of IBD.

According to the symptoms of IBD, the disruption of intestinal epithelial barrier was generally reported as the invasion of bacteria and pathogens. Thus, maintaining the integrity and tightness of the intestinal barrier is regarded as a strategy in the treatment of IBD [27]. As the major component of tight junction proteins, the Occludin and ZO-1 were demonstrated to be decreased in PC, whereas the intervention of GP had a relieving effect on that. In the protein-expression level, the dosage of GP-1 and GP-2 all had higher Occludin than the PC group. As revealed in the Caco-2 cells as well as the colitis mice, the treatment of bioactive peptides in foxtail millet protein hydrolysate (FMPH) increased intestinal ZO-1 and Occludin expressions compared with the DSS-induced groups [34]. In addition, the FMPH was involved in the NF-κB signal by inhibiting its phosphorylation process and then reduced the secretion of TNF-α and IL-6. In Caco-2 cell monolayers, the collagen peptides ameliorated intestinal epithelial barrier dysfunction by enhancing tight junction proteins of ZO-1 and occludin. Similarly, the NF-κB pathway was also inhibited by collagen peptides along with the suppressing effect on TNF-α secretion [35]. In the current study, the improvement on ZO-1 and Occludin proteins in GP treatment was the important clue for illustrating the mechanism of peptide-induced improving effects on IBD. Acted as the trigger of NF-κB, the higher expression of TLR-4 has been reported in IBD individuals [36]. Compared with DSS-induced mice, the supplement of GP suppressed the TLR-4 in mRNA and protein-expression levels, which implied that the NF-κB pathway would also be suppressed followed by the decreasing efficacy on TNF-α, IL-6, MCP-1, and IL-1β in serum. 

The gut microbiota has also been recognized to be associated with the generation of IBD. Commonly, the microbial diversity in IBD was declined along with the change of microbiota structure. In the current study, the relevant abundances of major gut microbiota were analyzed in phylum and gene levels. The *Akkermansia* is demonstrated to be a promising probiotic candidate for gut health and a decreasing abundance is revealed in IBD patients [29]. As expected, the GP intervention improved the *Akkermansia* abundance. In addition, the DSS treatment decreased the relative abundance of *Parasutterella*, which was improved by GP supplement. As a core component in humans and mice, *Parasutterella* has been defined to be correlated with bile acid maintenance and hypoxanthine metabolism [37]. In the DSS-induced colitis model, hypoxanthine modulated the energy balance of intestinal epithelium and exhibited an improving effect on intestinal barrier function [38]. Therefore, the elevated level of *Parasutterella* in GP treatment may exert beneficial effects on IBD as well as mucosal homeostasis. Furthermore, the intake of GP showed to decrease in the relative abundances of *Saccharibacteria*, *Bifidobacterium*, and *Peptococcus* than the DSS treatment. In the saliva microbiota dysbiosis from IBD patients, the *Saccharibacteria* along with another genus of *Absconditabacteria*, *Leptotrichia*, all were improved compared with healthy controls [39], whereas few studies have ever reported the change of *Saccharibacteria* in gut intestinal. GP supplement interferes with the abundance of *Akkermansia*, *Peptococcus*, *Bifidobacterium*, *Betaproteobacteria*, *Parasutterella*, *Saccharibacteria* along with the relieving effect on intestinal inflammatory cytokines, which was the first report about the intervention of collagen-derived peptides on IBD. 

The changes of the gut microbiota-induced metabolism function would be associated with the inflammatory responses and implied that the change of diet would inspire the microbial biological functions depending on the bidirectional communications among the host and gut microbiota. In the current study, the molecule weight of GP was less than 3 kDa, and most of the peptides were constituted by 7–15 amino acids. In vitro, the GP was treated with RAW264.7 macrophage cells with the investigation on its anti-inflammatory effect. Based on our knowledge, the treatment of GP showed suppressing effect on NO, TNF-α, IL-6, and IL-1βin the cells. However, during the intestinal digestion in vivo, the peptides chain would be destroyed by the digestive enzymes. Thus, the anti-inflammatory properties of GP in vitro could not indicate their activity once being digested and absorbed into the body. Thus, the animal trials were necessary to demonstrate the inflammatory regulating effect of GP. In the DSS induced colitis mice, the GP indeed exhibited a reduction on the inflammatory cytokines as well as improving the tight junctions of the colon. Known from the relative abundances change of gut microbiota, we speculated that some of the peptides in GP would escape from the digestive enzymes to reach the colon tissue. In addition, the peptides could also regulate the reactive oxygen species level in the gut and then exhibit the alleviating effect on the dysbiosis of gut homeostasis [40]. In general, the current research was the first study on the anti-inflammatory effect of bovine bone gelatin-derived peptides with their relieving functionality on IBD. According to the expression of the biomarkers in the colon, the dose of 300 mg/kg/day in GP-2 groups was much more effective than the GP-1 of 100 mg/kg/day. According to the study by Reagan, the dose in the animal trials could be converted to human equivalent dose by the body surface area normalization method [41]. Thus, the dose of 300 mg/kg/day in mice was converted to 24.32 mg/kg/day for humans (60 kg) and the potential clinical applicable dosage of GP was suggested to be 1459 mg/day. 

## 5. Conclusions

In summary, GP exert the anti-inflammatory effect in LPS-induced 264.7 microphage cells by suppressing the inflammatory cytokines. Meanwhile, the supplement of GP attenuated IBD symptoms and proinflammatory cytokines secretion along with resuming the tight-junction proteins in DSS-induced colitis mice. Moreover, the GP could regulate the structure of gut microbiota by promoting the relative abundances of *Akkermansia*, *Parasutterella*, and suppressing the relative abundances of *Peptococcus*, *Bifidobacterium* and *Saccharibacteria*. Despite that both assessments in cells preliminary and the following gavage test in vivo could not specifically illustrate which mechanism GP conducts, the beneficial function was interpreted. Thus, the GP may be regarded as a potential ingredient for ameliorating the inflammatory level in gut health and would be also developed as the multi-targeted functional foods for the treatment of IBD. 

## Figures and Tables

**Figure 1 nutrients-14-01479-f001:**
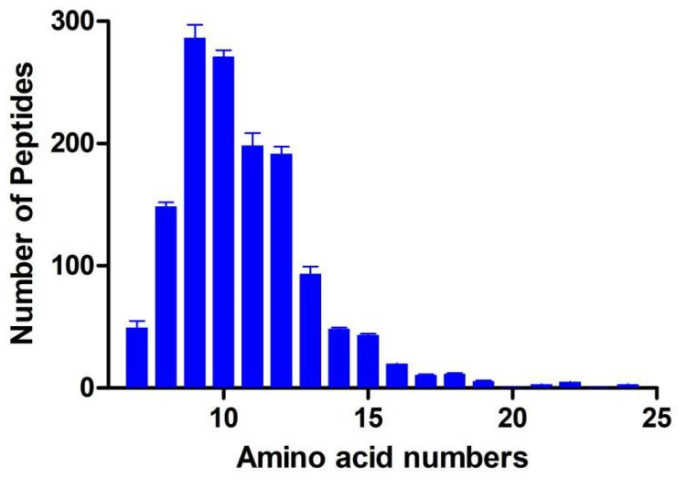
Distribution of peptides numbers in GP.

**Figure 2 nutrients-14-01479-f002:**
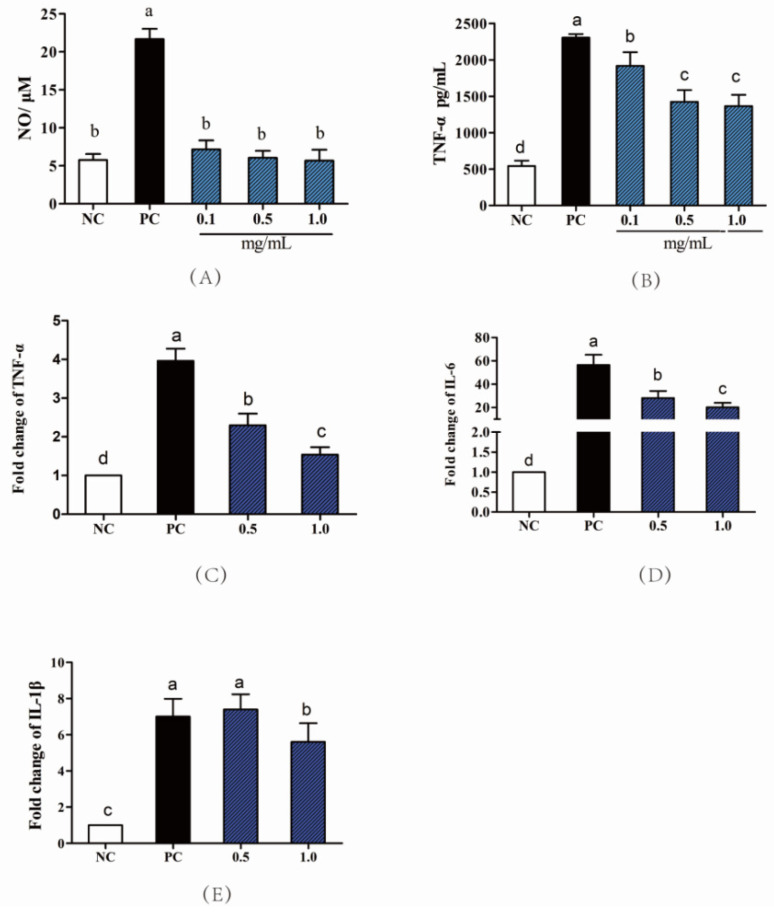
Anti-inflammatory activity of GP in RAW264.7 cells. (**A**) Effects of GP on the secretion of NO; (**B**) Effects of GP on the secretion of TNF-α; The mRNA expression of TNF-α (**C**), IL-6 (**D**), IL-1β (**E**) in LPS-stimulated RAW264.7 cells. NC means negative control; PC means positive control (LPS final concentration was 1 μg/mL); the GP was treated within concentration of 0.1, 0.5, 1.0 mg/mL. Different letters (a–d) represent significant differences (*p* < 0.05, *n* = 8) according to the one-way ANOVA (Tukey’s Multiple Comparison Test) analysis.

**Figure 3 nutrients-14-01479-f003:**
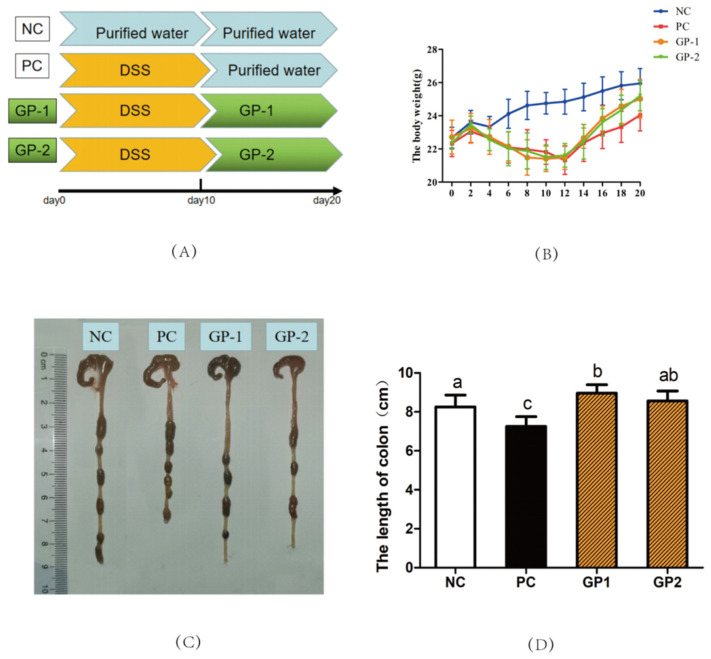
GP attenuated DSS-induced colitis in mice. Experimental design for the mice with DSS-induced colitis. The mice were assigned to four groups (**A**): NC, normal control; PC, positive control (1.5% DSS was supplied in water); GP-1 and GP-2, oral with 100 and 300 mg/kg/day respectively. (**B**) the change of body weight of mice during the trial. (**C**) Photographs of the colon. (**D**) Length of colon. All values represent the means ± SD (*n* = 8), and different letters represent significantly different of *p* < 0.05 according to the one-way ANOVA (Tukey’s Multiple Comparison Test) analysis.

**Figure 4 nutrients-14-01479-f004:**
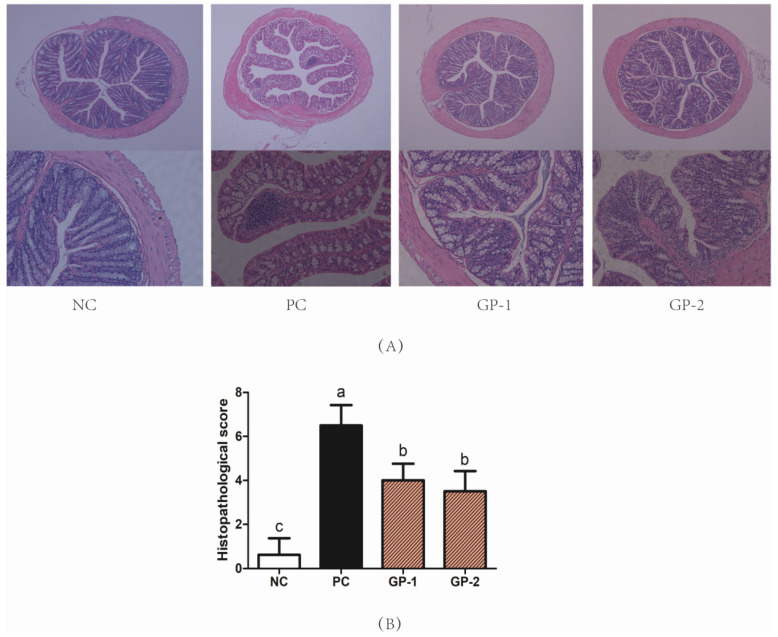
GP suppressed colonic tissue damage in DSS-induced colitis. The H&E sections (**A**) and histopathological scores (**B**) of colon tissue. DSS (1.5%) was supplied to mice in the water for 10 days. The H&E image (Bar = 100 μm) of colon samples were collected on Day 20. According to the one-way ANOVA (Tukey’s Multiple Comparison Test), the different letters mean significant difference of *p* < 0.05.

**Figure 5 nutrients-14-01479-f005:**
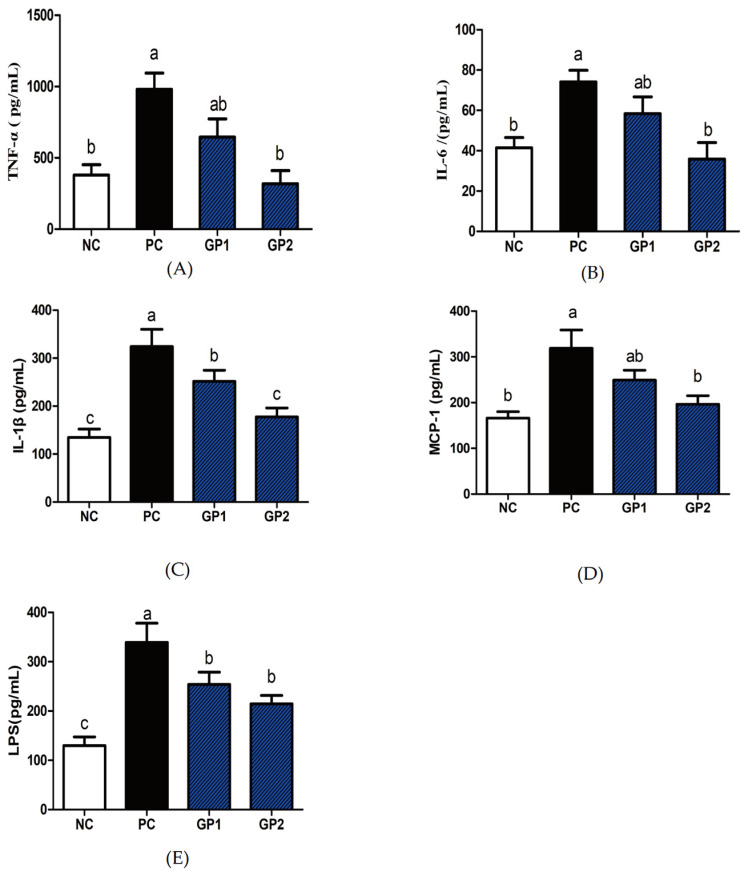
GP reduced inflammatory cytokines secretion in serum of DSS-colitis mice. The levels of TNF-α (**A**), IL-6 (**B**), IL-1β (**C**), MCP-1 (**D**), LPS (**E**) in serum were determined by ELISA. All values represent the means ± SD (*n* = 8). According to the one-way ANOVA (Tukey’s Multiple Comparison Test), the different letters mean significant differences of *p* < 0.05.

**Figure 6 nutrients-14-01479-f006:**
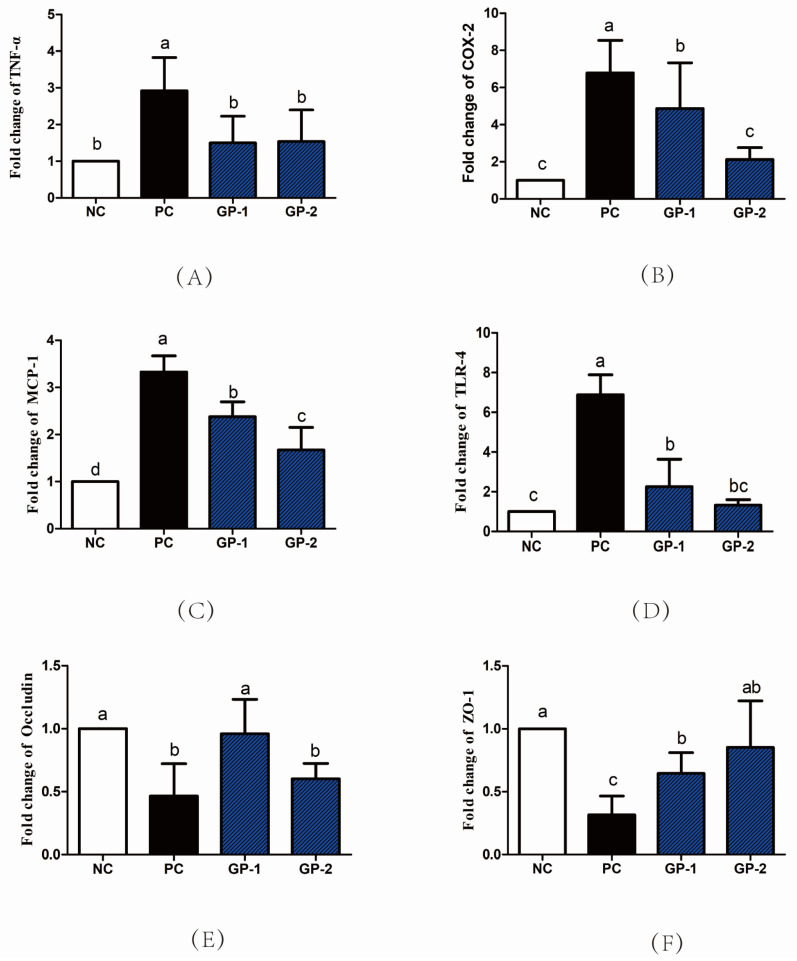
The mRNA expression levels of inflammatory cytokines and intestinal receptor proteins. The TNF-α (**A**), COX-2 (**B**), MCP-1 (**C**), TLR-4 (**D**), Occludin (**E**), ZO-1 (**F**) expression in colon tissues were measured by RT-PCR. All values represent the means ± SD (*n* = 8) and the differences among groups were analyzed by one-way ANOVA (Tukey’s Multiple Comparison Test). The different letters mean significant differences of *p* < 0.05.

**Figure 7 nutrients-14-01479-f007:**
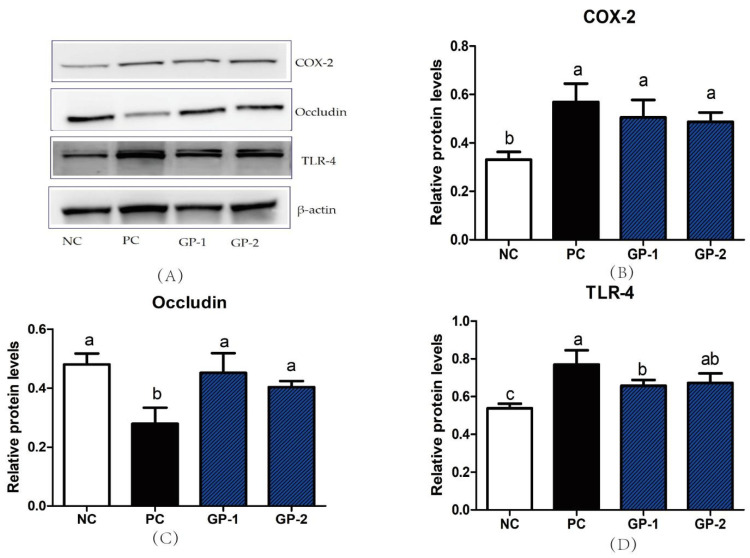
Western blot of proteins in colon. All the proteins were analyzed with the comparison of β-actin. The protein bands in Western blot analysis (**A**). The expression of COX-2 (**B**), Occludin (**C**) and TLR-4 (**D**). All values represent the means ± SD (*n* = 4) and the different letters mean significant differences of *p* < 0.05 by the analysis of one-way ANOVA (Tukey’s Multiple Comparison Test).

**Figure 8 nutrients-14-01479-f008:**
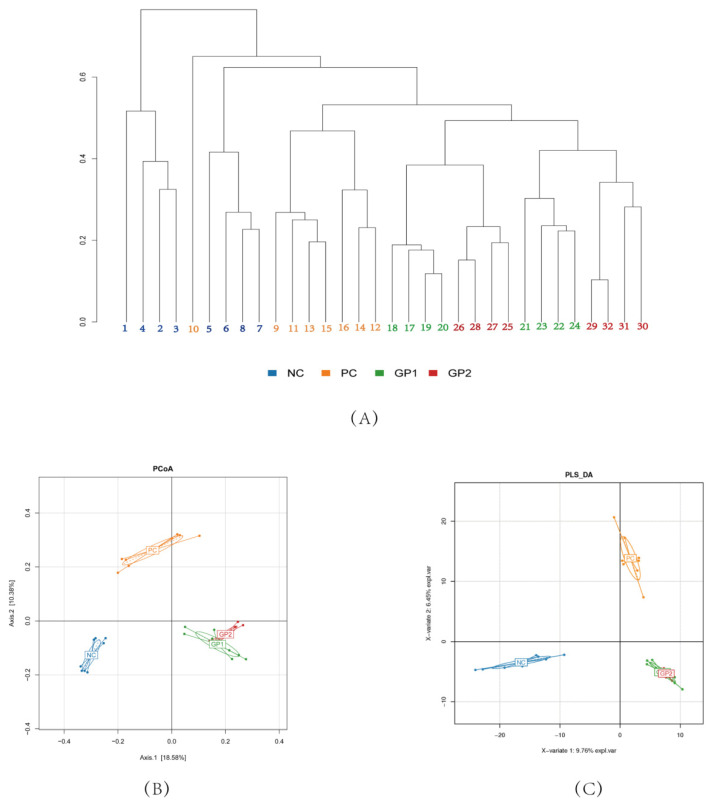
Effects of GP on microbial profiles based on OTU abundance. (**A**) Samples cluster tree; (**B**) PCA; (**C**) PLS.

**Figure 9 nutrients-14-01479-f009:**
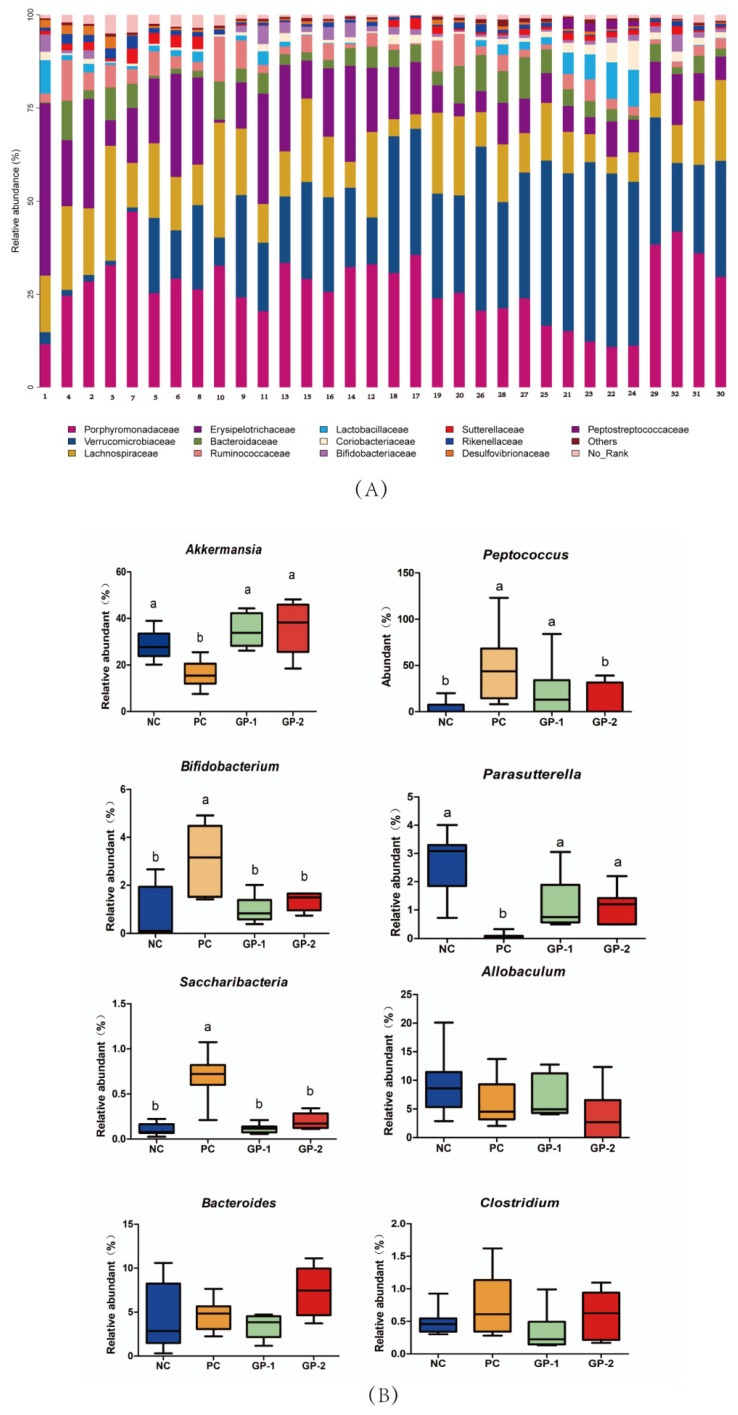
Gut microbial community structures in mice. (**A**) Microbial community bar plot by phylum level; (**B**) comparative analysis of relative abundances of gut microbiota at the gene level. The different letters represent significant differences between different groups (*p* < 0.05).

## Data Availability

Not applicable.

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
