# Peer review of "The Anti-Inflammatory Effect of Bovine Bone-Gelatin-Derived Peptides in LPS-Induced RAW264.7 Macrophages Cells and Dextran Sulfate Sodium-Induced C57BL/6 Mice"

_nutrients, 2022, doi:10.3390/nu14071479_

Round 1

Reviewer 1 Report

Reviewer' comments:

The manuscript ID: nutrients-1643761evaluates the anti-inflammatory role of bovin bone gelatin in reversing inflammation in both in vivo model of colitis and in vitro model of LPS-mediated macrophages inflammation.

This manuscript is scientifically interesting and need some ameliorations to fit the scope of the journal. The title and abstract are informative and give a clear idea of what to expect from the paper.

Some suggestions need to be addressed regarding the following points:

1.The Authors presents in vitro data on the therapeutic effects of this compound in an experimental model of LPS induced macrophages, without highlighting the rational of testing the effect of this compound on macrophages and not other cells of the intestinal tracts such as smooth muscle cells or epithelial cells. Please explain carefully this point in the introduction section.

  1. In both in vivo and in vitro models, authors measure the gene expression of both cytokines, inflammatory markers, protein junctions, and receptors by presenting RT-PCR results only without measuring the expression levels of inflammatory (COX-2, Monocyte chemoattractant protein-1) markers, protein junctions (Occludin, ZO-1) and receptors (TLR-4) activated in Colitis.
  2. It is important to measure the gene expression and provide the protein expression of these markers by doing Western blot analyses or immunofluorescence. Authors are highly encouraged to test by western blot the expression levels of COX-2, Occludin, TLR-4) and by Immunofluorescence the ZO-1, MCP-1), to confirm clearly the fact that this gelatin compound mediate therapeutic effects in colitis. Without doing these tests, the current findings are not sufficient and need the protein detection of these markers. Also, it is difficult to clearly state that this compound could reverse the colitis activated inflammation.

Author Response

  1. The Authors presents in vitro data on the therapeutic effects of this compound in an experimental model of LPS induced macrophages, without highlighting the rational of testing the effect of this compound on macrophages and not other cells of the intestinal tracts such as smooth muscle cells or epithelial cells. Please explain carefully this point in the introduction section.

Response:The bioactive peptides in bovine whey, oyster soft tissue, and sturgeon muscle were all demonstrated to have the anti-inflammatory effect on suppressing the expression of the inflammatory cytokines in RAW264.7 macrophages cells. In general, the RAW264.7 macrophages were primarily used as a cell model to evaluate the anti-inflammatory effect of bioactive compounds as its sensitivity to immune stimulation. In current study, we evaluated the inflammatory regualtion effect of GP in LPS induced RAW264.7 macrophages cells in the primary step, which as hoped to prove the anti-inflammatory effect. And based on the effect in vitro testing, we carried on the in vivo trials.

  1. In both in vivo and in vitro models, authors measure the gene expression of both cytokines, inflammatory markers, protein junctions, and receptors by presenting RT-PCR results only without measuring the expression levels of inflammatory (COX-2, Monocyte chemoattractant protein-1) markers, protein junctions (Occludin, ZO-1) and receptors (TLR-4) activated in Colitis. It is important to measure the gene expression and provide the protein expression of these markers by doing Western blot analyses or immunofluorescence. Authors are highly encouraged to test by western blot the expression levels of COX-2, Occludin, TLR-4) and by Immun of luorescence the ZO-1, MCP-1), to confirm clearly the fact that this gelatin compound mediate therapeutic effects in colitis. Without doing these tests, the current findings are not sufficient and need the protein detection of these markers. Also, it is difficult to clearly state that this compound could reverse the colitis activated inflammation. 

Response:Thanks for your suggestion and we have added the western blot results of cox-2, Occludin and TLR-4 so as to reveal the GP induced expression level of key peoteins. As shown in Figure 6, the expression of COX-2 in the DSS group was increased significnatly than the NC, and the GP supplementation suppressed COX-2 expression. In addition, the DSS induced inhibition on Occludin expression than NC, and the supplement of GP exhibited the inproving effect on that. Here, no significant differences was revealed in GP-1 and GP-2 groups. As for the TLR-4, the DSS also induced a suppressing expression than NC, whereas the GP had a relieving effect on it. In general, the supplement of GP had a sinificant effect on the key proteins, where the COX-2 and TLR-4 was suppressed and Occludin was inproved.

Figure 7. Western blot of proteins in colon. All the proteins were analysed with the comparison of β-actin. The protein bands in western blot analysisi (A). The expression of COX-2 (B), Occludin (C) and TLR-4 (D).

Reviewer 2 Report

The manuscript titled “The anti-inflammatory effect of bovine bone gelatin derived peptides in LPS-induced RAW264.7 macrophages cells and dextran sulfate sodium-induced C57BL/6 mice” explained that bone gelatin derived peptides has potency to attenuate endotoxin or DSS-inducible inflammation. The overall plot of the data set seems great; however, writing should be updated for the potential acceptance to Nutrients.

[Major comments]

  1. This manuscript is relatively lacking in the discussion. The majority of results are describing observation-based explanations; some discussion is available in the results section; however, it is limited. Therefore, the authors seriously consider providing a separate discussion section.
  2. The authors failed to provide the rationale to select bovine bone gelatin-derived peptides in RAW264.7 cells. In vitro experiments should be a supportive means for in vivo observations. Therefore, the treatment concentration should be determined/postulated by in vivo absorption rate through the gut. And, the treatment concentration should be implied in the circular vein (blood). Authors should provide a clear rationale to choose the treatment concertation of bovine bone gelatin-derived peptides in Raw264.7 cells besides cell viability assay.
  3. Generally, polypeptide passes the gut, it should be hydrolyzed as single amino acids or dipeptides. However, the authors directly treated 7-15 amino acids based bovine bone gelatin-derived peptides to RAW264.7 cells. Therefore, in vitro data is not practical and may be removed or transferred to a supplementary part. Once authors assert the in vitro data, authors should clearly demonstrate it is not practical.
  4. The authors should completely discuss how bovine bone gelatin-derived peptides attenuate systemically and gut inflammation. Does reduction of gut inflammation due to the protection of systemic inflammation?
  5. Potential clinical applicable dosage should be discussed.
  6. Authors need to have professional English correction services.

[Minor comments]

  1. Check all abbreviations.

- i.e. L107, etc

  1. Check all superscripts and subscripts.

- i.e. L101, 104, etc

  1. Formatting issues.

- i.e. L120, 122

  1. Coherence.

- Expressions; Figure vs Fig

- Font type; overall manuscript and figures

- y axis; Fig 2A NO/ uM?, Fig 5B IL-6/ ?

  1. No error bars

- Fig 1, Fig 2, Fig 6

  1. Statistical analytical methods are not listed in the figure captions.
  2. In Fig3D do NC and PC differ?
  3. Fig 4. There are only subjective data, please provide objective biological data.
  4. [L340] There are no intestinal receptor proteins.
  5. Outdated and limited references were cited. Please update.
  6. Please follow the Nutrients reference style using relevant software.

Author Response

  1. This manuscript is relatively lacking in the discussion. The majority of results are describing observation-based explanations; some discussion is available in the results section; however, it is limited. Therefore, the authors seriously consider providing a separate discussion section.

Response:Thanks for your suggestion and we have added the separate discussion part in the modified version.

Discussion

    It has been reported that DSS-induced colitis in mice had weight loss, diarrhea, hematochezia, and colon shorten symptoms. In the present study, the mice were checked with significant blood and diarrhea in the feces after 10 days of DSS treatment. As expected, the supplement of GP in two dosages could increase the weight loss along with the relieving effect on diarrhea and blood in the feces. As revealed in the scores of DAI, treatment with GP had a decreasing efficacy on the IBD symptoms. In addition, the abnormalities of inflammatory cytokines in serum, such as TNF-α, IL-6, IL-1β, and MCP-1, also played a vital role in the formation of IBD [31]. Compared with the DSS-induced group, the supplement of GP decreased secretion of inflammatory cytokines in serum. It is well known that the LPS is produced by Gram-negative bacteria with the effect of stimulating epithelial cells to increase COX-2 and then produce the intestinal inflammatory cascade [32]. Thus, the COX-2 can be regarded as a proinflammatory mediator and also a novel target participating in inflammation. In the current study, the change of COX-2 was investigated by expressions at the mRNA and protein levels, where the DSS induced a higher COX-2 expression and the GP had the relieving effect on that. TLR-4 is a signal of activating the NF-κB pathway with the trigger of further inflammatory cytokines secretion in IBD [33]. As been demonstrated in mRNA and protein levels, the TLR-4 checked to be elevated in PC and be further suppressed by GP treatment. In general, the regulation of these inflammatory factors is significant proof for the treatment of IBD.

According to the symptoms of IBD, the disruption of intestinal epithelial barrier was generally reported as the invasion of bacteria and pathogens. Thus, maintaining the integrity and tightness of the intestinal barrier is regarded as a strategy in the treatment of IBD [27]. As the major component of tight junction proteins, the Occludin and ZO-1 were demonstrated to be decreased in PC, whereas the intervention of GP had a relieving effect on that. In the protein expression level, the dosage of GP-1 and GP-2 all had higher Occludin than the PC group. As revealed in the Caco-2 cells as well as the colitis mice, the treatment of bioactive peptides in foxtail millet protein hydrolysate (FMPH) increased intestinal ZO-1 and Occludin expressions compared with the DSS induced groups [34]. In addition, the FMPH got involved in the NF-κB signal by inhibiting its phosphorylation process and then reduced the secretion of TNF-α and IL-6. In Caco-2 cell monolayers, the collagen peptides ameliorated intestinal epithelial barrier dysfunction by enhancing tight junction proteins of ZO-1 and occludin. Similarly, the NF-κB pathway was also inhibited by collagen peptides along with the suppressing effect on TNF-α secretion [35]. In the current study, the improvement on ZO-1 and Occludin proteins in GP treatment was the important clue for illustrating the mechanism of peptides-induced improving effects on IBD. Acted as the trigger of NF-κB, the higher expression of TLR-4 has been reported in IBD individuals [36]. Compared with DSS-induced mice, the supplement of GP suppressed the TLR-4 in mRNA and protein expression levels, which implied that the NF-κB pathway would also be suppressed followed by the decreasing efficacy on TNF-α, IL-6, MCP-1, and IL-1β in serum.

The gut microbiota has also been recognized to be associated with the generation of IBD. Commonly, the microbial diversity in IBD was declined along with the change of microbiota structure. In the current study, the relevant abundances of major gut microbiota were analyzed in phylum and gene levels. The Akkermansia is demonstrated to be a promising probiotic candidate for gut health and a decreasing abundance is revealed in IBD patients [29]. As expected, the GP intervention improved the Akkermansia abundance. In addition, the DSS treatment decreased the relative abundance of Parasutterella, which was improved by GP supplement. As a core component in humans and mice, Parasutterella has been defined to be correlated with bile acid maintenance and hypoxanthine metabolism [37]. In the DSS-induced colitis model, hypoxanthine modulated the energy balance of intestinal epithelium and exhibited an improving effect on intestinal barrier function [38]. Therefore, the elevated level of Parasutterella in GP treatment may exert beneficial effects on IBD as well as mucosal homeostasis. Besides, the intake of GP showed to decrease in the relative abundances of Saccharibacteria, Bifidobacterium, and Peptococcus than the DSS treatment. In the saliva microbiota dysbiosis from IBD patients, the Saccharibacteria along with another genus of Absconditabacteria, Leptotrichia, all were improved compared with healthy controls [39], whereas few studies have ever reported the change of Saccharibacteria in gut intestinal. GP supplement interferes with the abundance of Akkermansia, Peptococcus, Bifidobacterium, Betaproteobacteria, Parasutterella, Saccharibacteria along with the relieving effect on intestinal inflammatory cytokines, which was the first report about the intervention of collagen-derived peptides on IBD.

The changes of the gut microbiota-induced metabolism function would be associated with the inflammatory responses and implied that the change of diet would inspire the microbial biological functions depending on the bidirectional communications among the host and gut microbiota. In the current study, the molecule weight of GP was less than 3 kDa, and most of the peptides were constituted by 7-15 amino acids. During the intestinal digestion, the peptides chain would be destroyed by the digestive enzymes, which needs further study to demonstrate the bioavailability of GP. Known from the relative abundances change of gut microbiota, we speculated that some of the peptides in GP would escape from the digestive enzymes to reach the colon tissue. In addition, the peptides could also regulate the reactive oxygen species level in the gut and then exhibit the alleviating effect on the dysbiosis of gut homeostasis [40]. In general, the current research was the first study on the anti-inflammatory effect of bovine bone gelatin-derived peptides with their relieving functionality on IBD. According to the expression of the biomarkers in the colon, the dose of 300 mg/kg/day in GP-2 groups was much more effective than the GP-1 of 100 mg/kg/day. According to the study of Reagan, the dose in the animal trials could be converted to human equivalent dose by the body surface area normalization method [41]. Thus, the dose of 300 mg/kg/day in mice was converted to 24.32 mg/kg/day for humans (60 kg) and the potential clinical applicable dosage of GP was suggested to be 1459 mg/day.

  1. The authors failed to provide the rationale to select bovine bone gelatin-derived peptides in RAW264.7 cells. In vitro experiments should be a supportive means for in vivo observations. Therefore, the treatment concentration should be determined/postulated by in vivo absorption rate through the gut. And, the treatment concentration should be implied in the circular vein (blood). Authors should provide a clear rationale to choose the treatment concertation of bovine bone gelatin-derived peptides in Raw264.7 cells besides cell viability assay.

Response:In current study, the RAW264.7 cells was chosen to establish the LPS induced inflammatory model to investigate whether GP have inflammatory regulation function or not. The concentration in cells testing was referred to study of Gao et al [1]. Here, the dosage in vitro testing was not directly related to the dose in animal tests. Therefore, we just investigated the major inflammatory cytokines such as TNF-α, IL-6, IL-1β. Based on the results, the GP exhibited anti-inflammatory efficacy by suppressing the cytokines secretion, and then the animal trials were carried on to evaluate the effect of GP in mice colitis.

References

[1] Gao, R., Shu, W., Shen, Y., Sun, Q., Jin, W., Li, D., Yuan, L. Peptide fraction from sturgeon muscle by pepsin hydrolysis exerts anti-inflammatory effects in LPS-stimulated RAW264.7 macrophages via MAPK and NF-κB pathways. Food Sci. Hum. Well. 2021, 10, 110-118.

  1. Generally, polypeptide passes the gut, it should be hydrolyzed as single amino acids or dipeptides. However, the authors directly treated 7-15 amino acids based bovine bone gelatin-derived peptides to RAW264.7 cells. Therefore, in vitro data is not practical and may be removed or transferred to a supplementary part. Once authors assert the in vitro data, authors should clearly demonstrate it is not practical.

Response:The bioactive peptides in bovine whey, oyster soft tissue, and sturgeon muscle were all demonstrated to have the anti-inflammatory effect on suppressing the expression of the inflammatory cytokines in RAW264.7 macrophages cells. In general, the RAW264.7 macrophages were primarily used as a cell model to evaluate the anti-inflammatory effect of bioactive compounds as its sensitivity to immune stimulation. In current study, we evaluated the inflammatory regualtion effect of GP in LPS induced RAW264.7 macrophages cells in the primary step, which as hoped to prove the anti-inflammatory effect. And based on the effect in vitro testing, we carried on the in vivo trials.

  1. The authors should completely discuss how bovine bone gelatin-derived peptides attenuate systemically and gut inflammation. Does reduction of gut inflammation due to the protection of systemic inflammation?

Response:Add in the discussion part in line 494-499.

In the current study, the improvement on ZO-1 and Occludin proteins in GP treatment was the important clue for illustrating the mechanism of peptides-induced improving effects on IBD. Acted as the trigger of NF-κB, the higher expression of TLR-4 has been reported in IBD individuals. Compared with DSS-induced mice, the supplement of GP suppressed the TLR-4 in mRNA and protein expression levels, which implied that the NF-κB pathway would also be suppressed followed by the decreasing efficacy on TNF-α, IL-6, MCP-1, and IL-1β in serum.

  1. Potential clinical applicable dosage should be discussed.

Response:The applicable dosage has been discussed in the line 535-540.

According to the expression of the biomarkers in the colon, the dose of 300 mg/kg/day in GP-2 groups was much more effective than the GP-1 of 100 mg/kg/day. According to the study of Reagan, the dose in the animal trials could be converted to human equivalent dose by the body surface area normalization method. Thus, the dose of 300 mg/kg/day in mice was converted to 24.32 mg/kg/day for humans (60 kg) and the potential clinical applicable dosage of GP was suggested to be 1459 mg/day.

  1. Authors need to have professional English correction services.

Response:Following with your suggestion, the grammar and sentence of this article has been completely revised.

[Minor comments]

  1. Check all abbreviations.

- i.e. L107, etc

Response:Modified.

  1. Check all superscripts and subscripts.

- i.e. L101, 104, etc

Response:Modified.

  1. Formatting issues.

- i.e. L120, 122

Response: Modified.

  1.  

- Expressions; Figure vs Fig

- Font type; overall manuscript and figures

- y axis; Fig 2A NO/ uM?, Fig 5B IL-6/

Response:Modified.

  1. No error bars

- Fig 1, Fig 2, Fig 6

Response:Modifed.

  1. Statistical analytical methods are not listed in the figure captions.

Response:Added in the modified version.

  1. In Fig3D do NC and PC differ?

Response:Significant differences between groups have been relabeled in Fig 3D.

 Fig 3D. The length of colon

  1. Fig 4. There are only subjective data, please provide objective biological data.

Response: In the Fig.4, the H&E sections and histopathological scores of colon tissue were exhibited here, which was the subjective data. We did not have more objective data as this part the experiment was not planned. In the further IBD related trials, we would added the relevant objective biological datas.

  1. [L340] There are no intestinal receptor proteins.

Response:Added in modified version in line 323-329.

  1. Outdated and limited references were cited. Please update.

Response:In the revised manuscript, the references were updated with the deletion of outdated papers.

  1. Please follow the Nutrients reference style using relevant software.

Response:Modified in the revised version.

Round 2

Reviewer 2 Report

#3 were not completely answered. Authors should acknowlede potential limitation of their study in the discussion.

Author Response

#3 were not completely answered. Authors should acknowlede potential limitation of their study in the discussion. 

Response:Thanks for your suggestion. The discussion has been added in the line 527-537.

In vitro, the GP was treated with RAW264.7 macrophage cells with the investigation on its anti-inflammatory effect. Based on our konwn, the treatment of GP showed suppressing effect on NO, TNF-α, IL-6, and IL-1β in the cells. However, during the intestinal digestion in vivo, the peptides chain would be destroyed by the digestive enzymes. Thus, the anti-inflammatory properties of GP in vitro could not indicate their activity once being digested and absorbed into the body. Thus, the animal trials were necessary to demonstrate the inflammatory regulating effect of GP. In the DSS induced colitis mice, the GP indeed exhibited a reduction on the inflammatory cytokines as well as improving the tight jucntions of colon. Known from the relative abundances change of gut microbiota, we speculated that some of the peptides in GP would escape from the digestive enzymes to reach the colon tissue. 
